

# Sexual dimorphism in shell size of the land snail *Leptopoma perlucidum* (Caenogastropoda: Cyclophoridae)

Chee-Chean Phung, Ming-Huei Choo and Thor-Seng Liew

Institute for Tropical Biology and Conservation, Universiti Malaysia Sabah, Kota Kinabalu, Sabah, Malaysia

## ABSTRACT

Sexual dimorphism in the shell size and shape of land snails has been less explored compared to that of other marine and freshwater snail taxa. This study examined the differences in shell size and shape across both sexes of *Leptopoma perlucidum* land snails. We collected 84 land snails of both sexes from two isolated populations on two islands off Borneo. A total of five shell size variables were measured: (1) shell height, (2) shell width, (3) shell spire height, (4) aperture height, and (5) aperture width. We performed frequentist and Bayesian t-tests to determine if there was a significant difference between the two sexes of *L. perlucidum* on each of the five shell measurements. Additionally, the shell shape was quantified based on nine landmark points using the geometric morphometric approach. We used generalised Procrustes and principal component analyses to test the effects of sex and location on shell shape. The results showed that female shells were larger than male shells across all five measurements (all with $p$-values $< 0.05$), but particularly in regards to shell height and shell width. Future taxonomic studies looking to resolve the *Leptopoma* species' status should consider the variability of shell size caused by sexual dimorphism.

## INTRODUCTION

Molluscs exhibit a wide variety of sexual systems and strategies, with the superfamilies of gastropods being dioecious, simultaneously hermaphroditic, or sequentially hermaphroditic (*Collin, 2013*). Among these, terrestrial gonochoristic gastropods make up several superfamilies under the orders Architaenioglossa (clade Caenogastropoda) and Cycloneritida (clade Neritimorpha); the rest are primarily hermaphroditic. Additionally, there is little research on sexual dimorphism in shell forms of land snail superfamilies compared to freshwater and marine snail superfamilies.

Not all species within a genus exhibit sexual dimorphism (*Glöer, Albrecht & Wilke, 2007*; *Lazutkina et al., 2009*; *Riascos & Guzman, 2010*; *Andreeva et al., 2017*). Because shell size and shape are often used as diagnostic characteristics in species' taxonomy, it is important to understand the effects of sexual dimorphism on these traits. Sexual dimorphism can be observed in a shell's surface ornamentations (*Dutta et al., 2017*), size (*Jokinen, Guerette & Kortmann, 1982*; *Son, 1997*; *Estebenet & Cazzaniga, 1998*; *Glöer,*

Corresponding author
Thor-Seng Liew,
thorseng@ums.edu.my,
thorsengliew@gmail.com

*Albrecht & Wilke, 2007*; *Pastorino, 2007*; *Kurata & Kikuchi, 2000*; *Son & Hughes, 2000*; *Minton & Wang, 2011*; *Ng et al., 2019*; *Páll-Gergely et al., 2020*), shape (*Jokinen, Guerette & Kortmann, 1982*; *Keawjam, 1987*; *Son & Hughes, 2000*; *Minton & Wang, 2011*; *Márquez & Averbuj, 2017*), and pigmentation (*Gofas, 2001*; *Schilthuizen, Sipman & Zwaan, 2017*). The driving forces that cause dimorphisms in shell characteristics are generally related to reproductive selection (*Pastorino, 2007*; *Collin, 2018*).

To our knowledge, sexual dimorphism in shell size and shape of terrestrial caenogastropod shells has only been studied in five genera: *Cochlostoma*, *Obscurella*, *Plectostoma*, *Barnaia,* and *Streptaulus* (*Raven, 1990*; *Gofas, 2001*; *Schilthuizen et al., 2003*; *Reichenbach, Baur & Neubert, 2012*; *Páll-Gergely et al., 2020*). With the exception of *Plectostoma*, these species studied so far have elongate conical shells where female shells are more slender than males. *Schilthuizen et al. (2003)* found no differences in shape between the two sexes in the sinistroid shells of *Plectostoma*. In the present study, we investigate sexual dimorphism in the land snail—*Leptopoma perlucidum* (*De Grateloup, 1840*) (Cyclophoridea). The species exhibits high polymorphism in shell colour pattern (*Phung, Heng & Liew, 2017*) and it is not known whether this shell feature, as well as its size and shape, reflect sexual dimorphism. Therefore, we tested the differences of five linear measurements (*i.e.,* size) corresponding to the nine referenced landmarks (*i.e.,* shape) on the shell of male and female individuals.

## MATERIALS & METHODS

### Collection of specimens

We collected living *L. perlucidum* adults (shells with reflected lips), as snails without lips are likely to be immature from two islands, Tiga Island (N5°43′28″, E115°39′20″) and Gaya Island (N6°1′3″, E116°1′50″), both located at the west coast of Sabah, between January and August 2016. Sabah Parks approved this work (permit no. TS/PTD/5/4 Jld. 54 (112)). The snails were drowned until the animal's soft body fully extended so that the sexes could be determined and examined under a stereomicroscope as described by *Jonges (1980)* (Figs. 1B and 1C). A total of 84 *L. perlucidum* specimens were collected: nine females and 12 males from Gaya Island, and 33 females and 30 males from Tiga Island. All specimens were deposited in the *BORNEENSIS* collection at the Institute for Tropical Biology and Conservation, Universiti Malaysia Sabah (BOR/MOL 6651, 6653-6654, 7873-7875, 7877, 8739-8748, 8750-8754, 8756-8784, 8786-8794, 8797-8802, 8805-8809, 8813, 8816-8817, 9444-9447, 9449-9451, and 9726-9728).

### Shell morphology data

Each shell was photographed from the shell's aperture view using a Leica Image Analyzer M205. The shell size and shape measurements were obtained using the photographs. The shell size consisted of five linear measurements, measured to the nearest 0.1 mm: aperture height (AH), aperture width (AW), shell height (SH), shell width (SW), and spire height (SpH) (Fig. 1A). The shell shape was quantified using nine landmark coordinates corresponding to the dimensions of the different parts of the shell and recorded with ImageJ (Fig. 1A). The measurements and landmarks were selected based

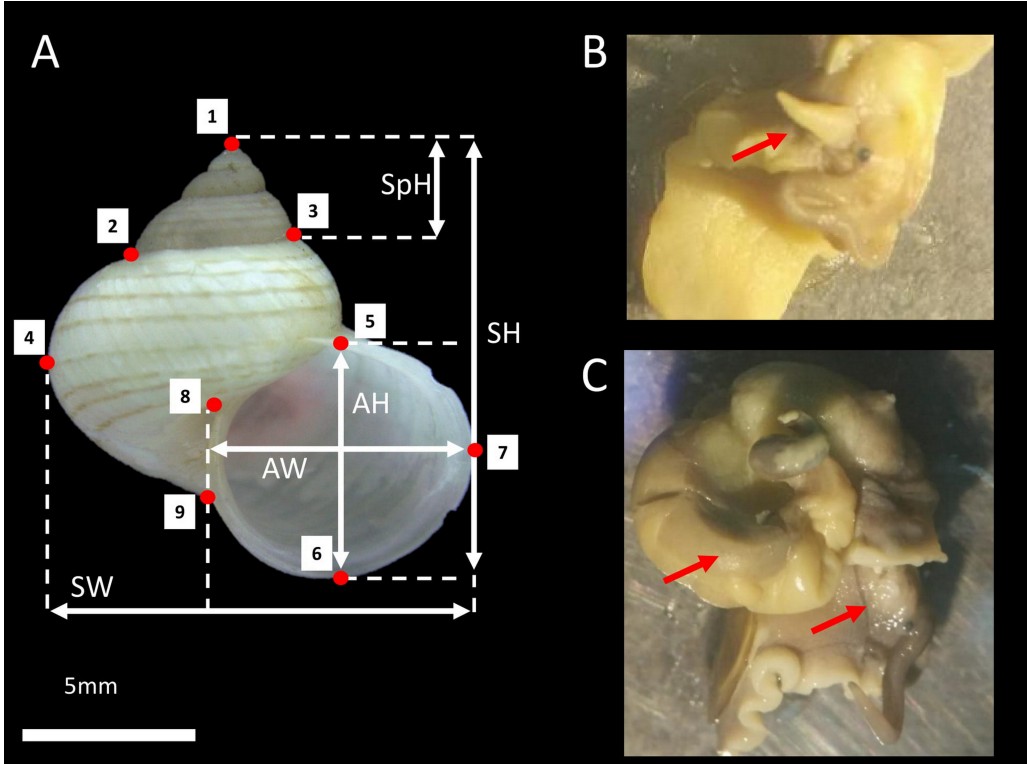

**Figure 1** *L. perlucidum* **shell and animals.** (A) Shell size and shape measurements for *L. perlucidum*. Shell sizes were measured in five ways: from the apertural view: SH, shell height; SpH, shell spire height; SW, shell width; AH, aperture height and AW, aperture width. Red points designated the nine landmarks for geometric morphometric. (B) Male specimen; red arrow indicated the penis. (C) The female specimen. The penis was not seen and the reproductive organ can be observed.

on the traits used for taxonomic purposes (*Vermeulen, 1999*; *Vermeulen & Liew, 2022*). Furthermore, these landmarks correspond to some of the landmarks used by *Larsson et al. (2020)* as developmentally descriptive shape parameters to simulate the three-dimensional logarithmic spiral growth of the shell.

## Data analysis
### Sexual dimorphism in shell size
We used two independent sample tests to determine whether there was a significant difference across each of the five shell measurements between both sexes of *L. perlucidum* for the two islands, respectively. No subset data (two locations *vs.* two sexes) violated the assumption of homogeneity of variances (Levene test) for all five shell measurements. However, three data subsets from Tiga Island (female AH, and male SW and SH) violated the normality assumption (Shapiro–Wilk test). Q-Q plots demonstrated that all five measurements had approximately normal distributed residuals. Because sampling was opportunistic and effect size (Cohen's d, *Cohen, 1988*) cannot be determined prior to sampling, we calculated effect size based on the differences between male and female snails in the five shell measurements for each location. The very large effect sizes found in the

measurements for both sexes on Tiga Island ranged from 0.75 to 1.20, while on Gaya Island, the values ranged from 1.15 to 1.92. Therefore, we checked the power of the tests after the experiment had been conducted by using the calculated effect size (d), significant level of 0.05, and the smaller n for each group, $n = 30$ and 9 for Tiga and Gaya Island respectively.

We conducted frequentist student's $t$-tests since the deviations from the normality assumption in these datasets were considered not too serious and we rejected the null hypothesis when $p < 0.05$. Simultaneously, we also performed Bayesian $t$-tests (BayesFactor—BF; *Morey & Rouder, 2015*) because the Bayesian framework could supplement the frequentist $p$-value (*Rouder et al., 2012*). Typically, $BF_{10} > 1$ is used to quantify evidence in favour of the alternative hypothesis. Ultimately, we based our conclusions on the inference of both frequentist ($p$-values) and Bayesian ($BF_{10}$) tests. All statistical tests and effect size calculations were performed using JASP software (version 0.12.2; *JASP Team, 2020*) while the power of the tests was assessed using the function "pwr.t.test" implemented in the 'pwr' packages for R (*Champely et al., 2020*).

***Sexual dimorphism in shell shape***
Shell shape was analysed using landmark-based geometric morphometrics. The digitised coordinate configurations of all specimens were subjected to Generalised Procrustes Analysis (GPA; *Rohlf & Slice, 1990*) to hold constant variations in size, orientation, and position. We then scaled them to standard unit centroid size. For the size comparison between the sexes, we did not use the centroid size of the set of landmarks, as the exact linear measurements were used in this study. GPA was done using the 'gpagen' function in the R package 'geomorph' (*Adams & Otárola-Castillo, 2013*). The aligned landmark coordinates produced by GPA were used as a set of shell shape variables for further analysis. First, principal component analysis (PCA) was applied using the 'plotTangentSpace' function in the same package to visualise the shape pattern variations. After that, we performed Procrustes ANOVA (regression for shape data) using the 'procD' function in the R package 'geomorph' to determine the effects of sex and location on shell shape. The analyses were performed in R (version 4.0.3; *R Core Team, 2015*). The completed R script can be found in File S1.

## RESULTS

### Sexual dimorphism in shell size
All five boxplots showed a difference across all shell size measurements between male and female *L. perlucidum* (Fig. 2). Both frequentist and Bayesian $t$-tests showed significant effect of sex on shell measurements, especially with respect to SH and SW (Table 1). The power of all the tests was above 0.8 for all shell measurements of both land snail populations, with the exception of AW for the Gaya Island population (power = 0.6). There were significant differences across all five measurements between female and male shells for both locations as revealed by the frequentist $t$-test (all with $p$-values <0.05) and the Bayesian $t$-test (BF inclusion >1). The female shell appeared to be larger than the male shell across all five measurements (Table 2, File S2).

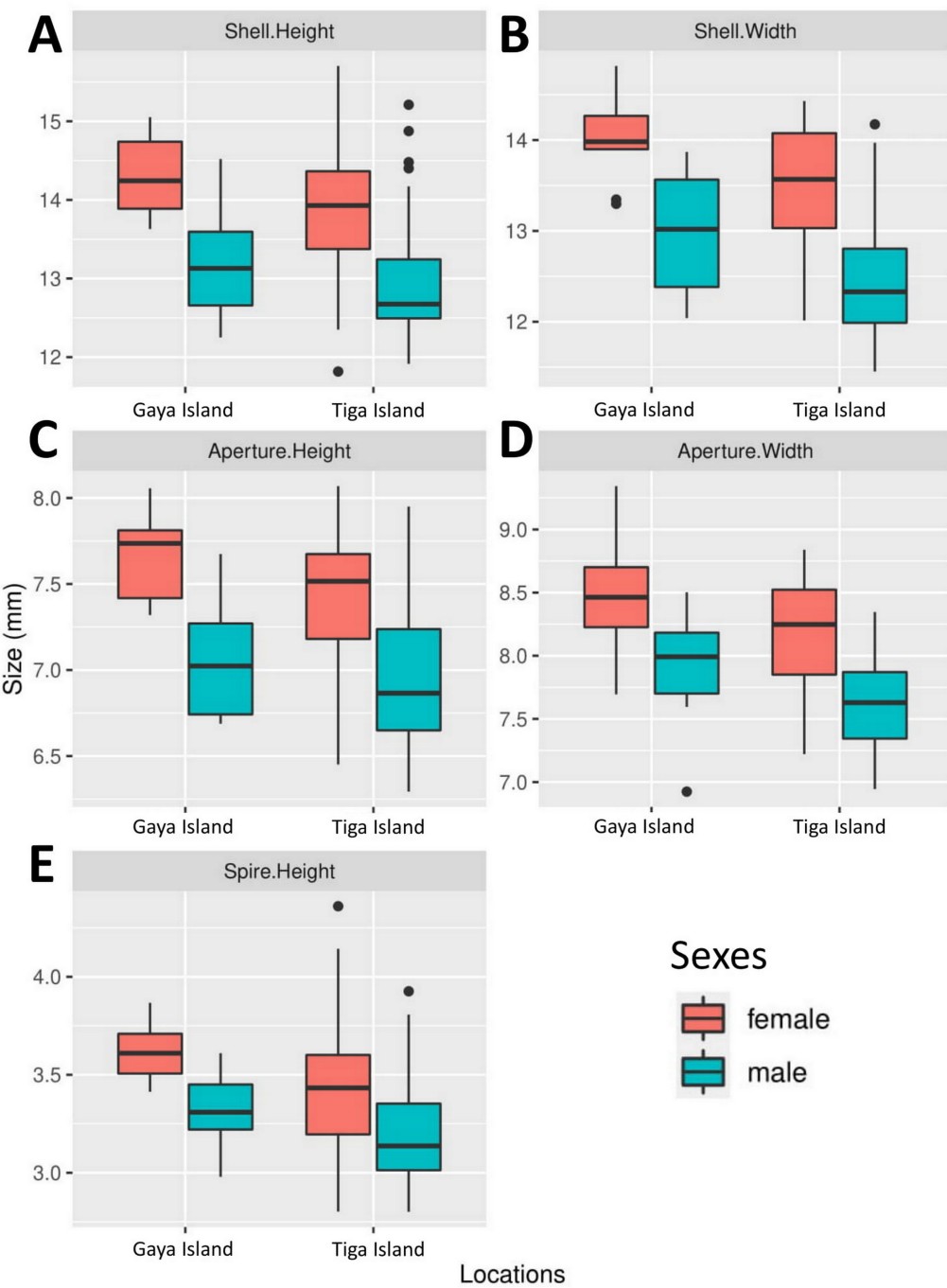

**Figure 2** **Boxplots show the differences in five quantitative shell measurements between specimens of opposite sex sampled from two locations.** Female, Gaya Island ($n = 9$); male, Gaya Island ($n = 12$); female, Tiga Island ($n = 33$); male, Tiga Island ($n = 30$). (A) Shell height. (B) Shell width. (C) Aperture height. (D) Aperture width. (E) Shell spire height.

**Table 1** Results of frequentist $t$-test, and Bayesian $t$-test for the effect of sexes and locations on five shell measurements.

| Measurements | Tiga Island | | | Gaya Island | | |
|---|---|---|---|---|---|---|
| | Frequentist $t$-test | Effect size | Bayesian $t$-test | Frequentist $t$-test | Effect size | Bayesian $t$-test |
| | $t$ | Cohen's d | $BF_{incl}$ | $t$ | Cohen's d | $BF_{incl}$ |
| Shell height | 3.941*** | 0.99 | 115.8 | 4.363*** | 1.92 | 72.5 |
| Shell width | 4.770*** | 1.20 | 1451.8 | 4.034*** | 1.78 | 39.4 |
| Aperture height | 3.716*** | 0.94 | 61.7 | 3.798** | 1.68 | 25.7 |
| Aperture width | 4.753*** | 1.20 | 1374.5 | 2.607* | 1.15 | 3.5 |
| Shell spire height | 2.971** | 0.75 | 9.3 | 3.928*** | 1.73 | 32.5 |

Notes.
Frequentist $t$-tests with significant $p$-values were annotated with asterisks. Significant $p$-values: * $<0.05$, ** $<0.01$, *** $<0.001$.

**Table 2** Summary of the shell measurements for the female and male *Leptopoma perlucidum* snails in two locations (Gaya Island and Tiga Island).

| Measurements (mm) | Gaya Island | | | Tiga Island | | |
|---|---|---|---|---|---|---|
| | Female | Male | Differences between female and male (%) | Female | Male | Differences between female and male (%) |
| | Mean ± S.D. | Mean ± S.D. | | Mean ± S.D. | Mean ± S.D. | |
| Shell height | 14.3 ± 0.5 | 13.2 ± 0.6 | +8.3% | 13.8 ± 0.9 | 13.0 ± 0.9 | +6.2% |
| Shell width | 14.0 ± 0.5 | 13.0 ± 0.6 | +7.7% | 13.5 ± 0.7 | 12.6 ± 0.8 | +7.1% |
| Aperture height | 7.7 ± 0.3 | 7.1 ± 0.4 | +8.5% | 7.4 ± 0.4 | 7.0 ± 0.5 | +5.7% |
| Aperture width | 8.5 ± 0.5 | 7.9 ± 0.4 | +7.6% | 8.2 ± 0.5 | 7.6 ± 0.4 | +7.9% |
| Shell spire height | 3.6 ± 0.1 | 3.3 ± 0.2 | +9.1% | 3.4 ± 0.4 | 3.2 ± 0.3 | +6.2% |

### Sexual dimorphism in shell shape

A total of 54.8% (PC1–31.8%, PC2–23%) of the shape variation in the nine-landmark coordinate dataset was explained by the first two principal components (Fig. 3). The thin-plate spline (TPS) transformation grids showed that the main source of the shape variations in the morphospace were from the aperture shape, *i.e.,* landmarks 5, 6, 7, 8, and 9. The scatterplot showed significant overlaps between the sex and location of *L. perlucidum* in their shell shape. There were no significant differences between the shell shapes of male and female *L. perlucidum* from the two different locations (Procrustes ANOVA—Sexes: $df = 1$, $F = 1.5247$, $p = 0.150$; locations: $df = 1$, $F = 1.6907$, $p = 0.117$; sex and location interaction: $df = 1$, $F = 0.4359$, $p = 0.879$) (File S3). Shell images of all specimens used in this study can be found in File S4.

## DISCUSSION

### Sexual dimorphism in *L. perlucidum* shell form

This is the first investigation into sexual dimorphism in shell size and shape in the Cyclophoridae. This species exhibits sexual dimorphism in shell size but not in shell shape. Females have larger shells than males despite shell size variations. To date, only a handful of land snail studies have investigated the size variations between both sexes, and all studies showed that females are more slender and larger than males (*Gofas, 2001*; *Reichenbach, Baur & Neubert, 2012*; *Páll-Gergely et al., 2020*). Although females can be at least 6% ($\sim$1mm)

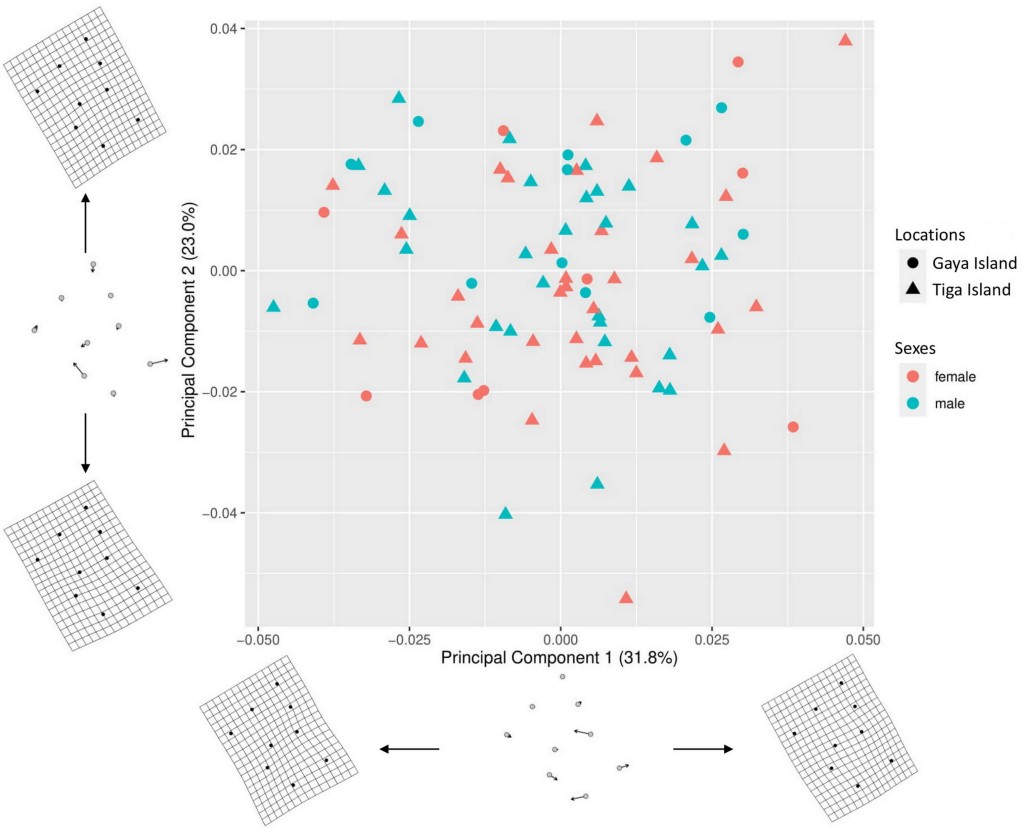

**Figure 3** **Principal component analysis (PCA) for shell shape of *L. perlucidum*.** Morphospace is represented by a scatterplot of the scores from the first two principal components of the geometric morphometric analysis of the nine shell landmarks. Thin-plate spline (TPS) transformation grids represent shape variation at the extremes of the two principal component axes.

larger than males in the same *L. perlucidum* population, this may not be noticeable even when comparing the shells of both sexes side by side. *Páll-Gergely et al. (2020)* pointed out that these differences can only be revealed when rigorous statistical methods are used, as is the case in most other studies.

On the other hand, there is an abundance of sexual dimorphism studies on the shell form of aquatic and marine snails. These studies overwhelmingly show that females are larger than males (*Jokinen, Guerette & Kortmann, 1982*; *Son, 1997*; *Estebenet & Cazzaniga, 1998*; *Kurata & Kikuchi, 2000*; *Son & Hughes, 2000*; *Minton & Wang, 2011*; *Ng et al., 2019*). However, there has been a few exceptions in which males were larger than females (*Kurata & Kikuchi, 2000*) or where there was no sexual dimorphism in shell size (*Tripoli et al., 2015*; *Vaux et al., 2017*). In addition to the overall shell size in terms of total height and width, some studies also showed that females had a larger aperture size compared to males (*Cazzaniga, 1990*; *Son, 1997*; *Minton & Wang, 2011*), or that the shells of females were more globose than males' (*Jokinen, Guerette & Kortmann, 1982*; *Keawjam, 1987*; *Son & Hughes, 2000*; *Minton & Wang, 2011*).

Currently, the biology, specifically the mating process, of the *Leptopoma* species remains unexplored. Therefore, we can only hypothesise that females have larger shells than males because of the different morphologies of their reproductive systems. In studies on other taxa, it has been hypothesised that sexual dimorphism in caenogastropod shells (females either have a larger shell size or a more globose shell) appears to be primarily related to female fecundity or egg mass production (*Collin, 2018*). Females generally have larger sized shells, especially in the last whorl, resulting in a higher shell volume for egg deposition (*Gofas, 2001*).

Studies on other taxa found that the driving forces for sexual dimorphism in shell form include sexual selection and the two sexes' different growth rates (*Jokinen, Guerette & Kortmann, 1982*; *Kurata & Kikuchi, 2000*; *Riascos & Guzman, 2010*). The underlying driving forces of sexual dimorphism in the shell size of *L. perlucidum* need to be similarly investigated in future studies with more field observations and experiments to determine whether females are larger than males in mating pairs (*e.g.*, *Ng et al., 2019*) and the growth rate of both sexes.

### The implications to taxonomic studies on the genus *Leptopoma*

Future taxonomic research to resolve the status of the *Leptopoma* species would need to take into account the variability of shell size caused by sexual dimorphism and geographical variations (*Phung, Heng & Liew, 2017*). There are 125 valid names for *Leptopoma* species, and there are another 154 invalid names due to various nomenclature reasons (*MolluscaBase, 2021*). The last intensive review, conducted by *Kobelt (1902)* on 105 *Leptopoma* is outdated, and subsequent taxonomic works have mainly dealt with a handful of species in relatively small geographic areas (*Gude, 1921*; *Vermeulen, 1999*; *Phung, Heng & Liew, 2017*). There has been no quantitative comparison of shell shapes among species, except *Phung, Heng & Liew (2017)*. The species diversity hotspot is in the Philippines archipelago, where many species occur sympatrically over thousands of islands (*GBIF, 2021*). Our study shows that in *L. perlucidum* at least, no shape differences (which would further complicate a taxonomic revision of the genus) are found between males and females.

## ACKNOWLEDGEMENTS

We thank Justinus Guntabid, Sukur B. Sukardi, Muhammad Aliff B. Suhaimin, Simon Limbawang, Victor Siam, and many other Sabah Parks staff members for providing logistical support throughout our fieldwork. We also appreciate the assistance from Yeong Kam Cheng, Simon Kuyun, Foo She Fui, Phung Kin Wah, and Jasrul Dulipat in sampling land snails on the islands. We thank Menno Schilthuizen and two anonymous reviewers for the constructive comments.

### Funding

This work was supported by the Fundamental Research Grant Scheme, Ministry of Higher Education, Malaysia (No. RAGS/1/2015/WAB13/UMS/02/1; UMS RAG0063-STWN-2015). The funders had no role in study design, data collection and analysis, decision to publish, or preparation of the manuscript.

### Grant Disclosures

The following grant information was disclosed by the authors:
Fundamental Research Grant Scheme, Ministry of Higher Education, Malaysia: RAGS/1/2015/WAB13/UMS/02/1, UMS RAG0063-STWN-2015.

### Competing Interests

The authors declare there are no competing interests.

### Author Contributions

- Chee-Chean Phung conceived and designed the experiments, performed the experiments, analyzed the data, authored or reviewed drafts of the article, and approved the final draft.
- Ming-Huei Choo conceived and designed the experiments, performed the experiments, authored or reviewed drafts of the article, and approved the final draft.
- Thor-Seng Liew conceived and designed the experiments, performed the experiments, analyzed the data, prepared figures and/or tables, authored or reviewed drafts of the article, and approved the final draft.

### Field Study Permissions

The following information was supplied relating to field study approvals (*i.e.*, approving body and any reference numbers):

The research was conducted in Tiga Islands and Tunku Abdul Rahman Marine Parks with the permission of Sabah Parks (Permit No. TS/PTD/5/4 Jld. 54 (112)).

### Data Availability

The raw data is available in the Supplementary Files.

### Supplemental Information

Supplemental information for this article can be found online at http://dx.doi.org/10.7717/peerj.13501#supplemental-information.

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
