# Peer review of "Sexual dimorphism in shell size of the land snail Leptopoma perlucidum (Caenogastropoda: Cyclophoridae)"

_PeerJ, doi:10.7717/peerj.13501_

## Round 0.1 · original submission · Major Revisions

Dear Drs. Phung and colleagues:

Thanks for submitting your manuscript to PeerJ. I have now received three independent reviews of your work, and as you will see, the reviewers raised some concerns about the research. Despite this, these reviewers are quite optimistic about your work and the potential impact it will have on research studying sexual dimorphism in land snails. Thus, I encourage you to revise your manuscript, accordingly, taking into account all of the concerns raised by the three reviewers.

Please ensure that your figures and tables contain all of the information that is necessary to support your findings and observations.

Please edit the manuscript for clarity and typos. There appear to be some key references missing. Your coverage of the literature on sexual dimorphism in snails was considered scant by the reviewers.

It seems the manuscript can benefit from restructuring, with the intended objectives clearly defined and addressed throughout. Areas in need of improvement include your taxonomic assessment and hypotheses for the observed phenotypes.

Good luck with your revision,

-joe

·

Basic reporting

The title is not very clear; I would suggest "Sexual dimorphism in shell size of the land snail Leptopoma perlucidum (Caenogastropoda: Cyclophoridae)" (The fact that this study was carried out in Sabah is probably not sufficiently relevant to be in the title; only differences in size were found, not in shape; and it seems more logical to mention the shell size immediately after "sexual dimorphism" and not only after the species name.)

In the Abstract, "islands in Borneo" should be changed to "islands off Borneo"

I noticed numerous stylistic and linguistic details in the text that could be improved during the editing process.

The Introduction should be supplemented with a brief paragraph in which the distribution of gonochorism and hermaphroditism in the Mollusca is outlined, because this limits the taxa in which research on sexual dimorphism is possible.

In the Introduction (L. 82), it says: "To our knowledge, sexual dimorphism in terrestrial caenogastropod shells has only been studied in four genera: Cochlostoma, Obscurella, Barnaia, and Streptaulus." However, we also studied it in the genus Plectostoma:

>>for colour (Sexual dimorphism in shell coloration of Plectostoma (Caenogastropoda: Diplommatinidae) is caused by polyenes; M Schilthuizen, I Sipman, H Zwaan - Journal of Molluscan Studies, 2018)

>>for shape (Schilthuizen, M., Rosmaineh B. Rosli, Abdul Muji B. Mohd. Ali, M. Salverda, H. van Oosten, H. Bernard, M. Ancrenaz & I. Lackman-Ancrenaz, 2003. The ecology and demography of Opisthostoma (Plectostoma) concinnum s.l. (Gastropoda: Diplommatinidae) on limestone outcrops along the Kinabatangan River. In: Kinabatangan Scientific Expedition (Maryati Mohamed, B. Goossens, M. Ancrenaz & M. Andau, eds.). UMS, Kota Kinabalu, Malaysia.

I don't understand how sexual dimorphism can result from "selection pressures among different distant geographical populations" (L. 69-70).

Experimental design

In L. 124 it says, "the ANOVA is considered a robust test against the normality assumption (Zar, 1999)". I presume what is meant is that the ANOVA is considered robust against violations of the normality assumption.

In the description of the geometric morphometrics, the package 'plotTangentSapce' is said to have been used (L. 140). Wouldn't this be 'plotTangentSpace'?

In L. 96-97, it might be good to add that the sexual anatomy of L. perlucidum has been described by Jonges (1980; Bijdr. Dierk.)

Validity of the findings

In L. 184, it says, 'This study of L. perlucidum sexual dimorphism in shell form is the first investigation of land snails in the family Cyclophoridae'. This is somewhat misleading since to the uninitiated reader it may suggest that no studies in Cyclophoridae have ever been conducted. I would suggest instead: 'This is the first investigation into sexual dimorphism in shell size and shape in the Cyclophoridae.'

The last few sentences of the discussion do not seem very appropriate, since this study is mainly about sexual dimorphism, not about geographic variation. I would suggest to end on a positive note, namely that this study shows that, in L. perlucidum at least, no shape differences (which would further complicatie a taxonomic revision of the genus) are found between males and females.

Reviewer 2 ·

Basic reporting

1. Basic reporting
Being non-native English, I nevertheless commented on some sentences (see 4 below). This is a mixture of linguistics and biology.
The context is clearly indicated.

Experimental design

2. Experimental design
Unfortunately, my apologies, I do not know that scope sufficiently well.
The research question is well defined. To some extent there is a knowledge gap, next to many similar gaps for other taxa.
The methods are clearly described.

Validity of the findings

3. Validity of the findings
There is no novelty for gastropods, but only for a particular genus. The findings are not conspicuously different from what we know for other genera that have been investigated. The impact will be meagre.
I feel not competent to judge the statistics. I fully rely on the corresponding author.
The conclusions are clear, correct, and linked to the research question.

Additional comments

4. Comments, questions of the referee
42 The "effect" of a location is something dubious. Drift is possible equally well.
44 "the between"??
46 Geographical variation is a very well known phenomenon for every taxonomist.
67 Adapt wording
69 Is sexual dimorphism a selection pressure?
72 Adapt wording since there is no geographical dimorphism
83 "Both"?
92 Explain why "reflected lips" is relevant and adapt the wording since without such a lip the snail will probably be immature.
221-223 This applies to all gastropods with potential sexual dimorphism, not only Leptopoma. This is well known among taxonomists.

Reviewer 3 ·

Basic reporting

There are many studies of sexual dimorphism using mollusks as subjects, dating back to the 19th century.

A quick Google Scholar search revealed papers on the sexual dimorphism in Plectostoma (Schilthuizen et al. 2017, cited in the paper), Arinia (Maassen 2008), and Chondrina (Baur et al. 1993) as well. Obviously a more thorough search is needed.

Experimental design

The sample sizes from Gaya Island are not large enough to support the significant findings reported in the paper. Using a power analysis (80%), at least 13 individuals from each sex needed to be used to achieve significance at p<0.05.

There a better and more thorough ways to analyze shell outlines that with a set of nine landmarks, including the addition of more landmarks, curve capture, and Fourier analysis. A more robust analysis of shell shape is warranted, especially when looking for potentially subtle differences between sexes.

In addition to shell shape, geometric morphometrics can assess shell size through analysis of centroid size if the landmarks are scaled appropriately to start.

Validity of the findings

The sample sizes from Gaya Island are not large enough to support the significant findings reported in the paper (see power analysis comment under Experimental design).

Given the small number of landmarks, it is unclear whether sexual shape dimorphism is actually present or not.

Additional comments

The discussion says very little other than that females may be larger than males in certain populations of this species. No explanation nor proposed reason for this difference is provided. The discussion regarding the taxonomy of the genus and the number of nominal species lacks context – the reader doesn’t know how many species are based on size alone, what the overlaps of the species are, etc.

---

## Round 0.2 · Minor Revisions

Dear Drs. Phung and colleagues:

Thanks for revising your manuscript. The reviewers are very satisfied with your revision (as am I). Great! However, there are a few minor edits to make per reviewer 3. Please address these ASAP so we may move towards acceptance of your work.

Best,

-joe

·

Basic reporting

The revision has addressed all the points raised by myself and other reviewers in a satisfactory manner. I have no further comments.

Experimental design

The revision has addressed all the points raised by myself and other reviewers in a satisfactory manner. I have no further comments.

Validity of the findings

The revision has addressed all the points raised by myself and other reviewers in a satisfactory manner. I have no further comments.

Additional comments

The revision has addressed all the points raised by myself and other reviewers in a satisfactory manner. I have no further comments.

Reviewer 2 ·

Basic reporting

Sexual dimorphism in shell size of the land snail Leptopoma perlucidum (Caenogastropoda: Cyclophoridae)
The authors have taken the comments of the reviewers seriously and have considerably adapted the manuscript at several relevant points.
I see no way for additional improvement.
Edmund Gittenberger

Experimental design

--

Validity of the findings

--

Additional comments

--

Reviewer 3 ·

Basic reporting

No comment.

Experimental design

No comment.

Validity of the findings

No comment.

Additional comments

The revisions that the authors have made in this version have improved the manuscript significantly. I have included some minor editorial changes in the text.

Annotated reviews are not available for download in order to protect the identity of reviewers who chose to remain anonymous.

---

## Round 0.3 · accepted · Accept

Dear Drs. Phung and colleagues:

Thanks for revising your manuscript based on the concerns raised by the reviewer. I now believe that your manuscript is suitable for publication. Congratulations! I look forward to seeing this work in print, and I anticipate it being an important resource for groups studying sexual dimorphism in land snails. Thanks again for choosing PeerJ to publish such important work.

Best,

-joe